# Optimization of Lipid Extraction from *Spirulina* spp. by Ultrasound Application and Mechanical Stirring Using the Taguchi Method of Experimental Design

**DOI:** 10.3390/molecules27206794

**Published:** 2022-10-11

**Authors:** Emilia Neag, Zamfira Stupar, Cerasel Varaticeanu, Marin Senila, Cecilia Roman

**Affiliations:** National Institute for Research and Development of Optoelectronics Bucharest INOE 2000, Research Institute for Analytical Instrumentation, 67 Donath Street, 400293 Cluj-Napoca, Romania

**Keywords:** ANOVA, lipids, orthogonal array, signal-to-noise ratio, Taguchi method

## Abstract

The present study uses the Taguchi method of experimental design to optimize lipid extraction from *Spirulina* spp. by ultrasound application and mechanical stirring. A Taguchi *L*_9_ orthogonal array was used to optimize various parameters, such as methanol: chloroform (M:C) ratio, biomass: solvent ratio, and extraction time for lipid extraction. The results were analyzed using the signal-to-noise (*S*/*N*) ratio and analysis of variance (ANOVA). The biomass: solvent ratio significantly influenced lipid content (*p* < 0.05) with 92.1% and 92.3% contributions to the lipid and *S*/*N* ratio data, respectively. The extraction time presented a contribution value of 5.0%, while the M:C ratio presented the most negligible contribution of 0.4% for *S*/*N* data. The optimum extraction conditions were: M:C ratio of 1:1, biomass: solvent ratio of 1:60, and extraction time of 30 min. The predominant fatty acids were palmitic acid (44.5%), linoleic acid (14.9%), and gamma-linolenic acid (13.4%). The confirmation experiments indicated a lipid content of 8.7%, within a 95% confidence interval, proving the Taguchi method’s effectiveness in optimizing the process parameters for lipid extraction.

## 1. Introduction

Microalgae produce various high-value compounds, such as pigments, polysaccharides, fatty acids, and vitamins [1]. In addition, *Spirulina* spp. is used as a dietary supplement due to its active ingredients that positively influence human health [2].

Lipids play an important role in offering a green alternative energy source. Their production depends on microalgae species and culture conditions, such as nutrients, salinity, light intensity periods, temperature, pH, and the association with other microorganisms. Usually, the total lipids in microalgae represent 20% to 50% of total biomass in dry weight (DW) [1,3,4]. It has been reported that *Spirulina* spp. has a total lipid content in the range of 6.4–14.3% DW [5,6].

The successful commercialization of lipid extraction on a larger scale depends on the microalgae growth optimization, efficient lipid extraction, and their transesterification to biodiesel. The solvent extraction methods are the most commonly used for lipid extraction due to the higher lipid recovery yields [7]. Various solvents have been used for lipid extraction, such as n-hexane, ethanol, 1-butanol, and dimethyl ether. In addition, mixtures of solvents, such as chloroform/methanol, n-hexane/ethanol, n-hexane/isopropanol, and acetone/dichloromethane, have been studied, out of which the most common solvent used for lipid extraction is chloroform/methanol due to its fast extraction [8].

The mechanical methods are environmentally friendly and cheap but lead to lower lipid recovery yields and the possible degradation of lipids. Few studies have reported the development of solvent-free extraction methods for algal biomass, such as the osmotic pressure and isotonic extraction method. Combining enzymatic and mechanical/solvent extraction methods can be a promising alternative to extract lipids from microalgae biomass efficiently. Enzyme-assisted extraction combined with another method can reduce solvent and energy consumption or increase lipid recovery yields [7].

Ultrasound treatment has gained interest in recent years due to its advantages, such as improved lipid extraction yield and kinetics, low consumption of solvents, short extraction time, low economic and environmental impacts, few instrumental requirements, and easy upscalability compared to conventional methods [9,10,11]. The findings proved the effectiveness of the ultrasound-assisted extraction using an ultrasound bath and ultrasound probe for lipid extraction from microalgae [10]. Fattah et al. reported that ultrasonication was found to be the best method to extract the lipids from *Spirulina* spp., with a percentage oil yield of 6.6% as compared with Soxhlet and solvent extraction processes [6]. Pohndorf et al. obtained a lipid content of 5.8 ± 0.6 g/100 g after extraction by the cold method, using polar solvents (chloroform:methanol 2:1 *v*/*v*) [12]. Liu et al. studied pilot-scale lipid extraction from *Spirulina* spp. using supercritical CO_2_, adding methanol as a co-solvent. The results showed that under an operating pressure of 4000 psi, the lipid extraction yield for *Spirulina* spp. increased by 80% [13]. Thus, it is important to identify the optimum conditions for lipid extraction from an economic point of view. The Taguchi method is simple and efficient and uses an experimental design called an orthogonal array to determine the optimum parameters with a small number of experiments by minimizing the interference of uncontrolled factors [14,15]. The signal-to-noise (*S*/*N*) ratio proposed by Taguchi is useful for finding the optimum levels of the parameters. The reproducibility and stability of a studied process increase while the fluctuations are minimized [16]. A high S/N ratio suggests optimum quality with minimum variation [17]. The S/N ratio is used to quantitively measure a response (e.g., lipid content) [18]. The performance characteristics, namely lower-the-better, larger-the-better, and nominal-the-better, are used to calculate the *S*/*N* ratio in Taguchi’s methodology [14,19]. According to the purpose of the experiments, the lower-the-better equation is used to minimize the system response (quality decreases as the system response increases), and the larger-the-better equation is used to maximize the system response (quality increases as the system response increases). The nominal-the-better equation is used to achieve a particular target value [20]. As stated by Fattah et al., the maximization of the output from feedstocks with low lipid content has not been studied considerably and should be addressed [6].

The present study aimed to maximize the lipid extraction from *Spirulina* spp. by optimizing various parameters that have an impact on the lipid extraction process.

## 2. Results and Discussion

### 2.1. Selection of Optimal Levels and Estimation of Optimum Response Characteristics

The extraction of lipids from *Spirulina* spp. after ultrasonic and mechanical stirring was evaluated. According to the literature, ultrasound is one of the most promising pretreatment methods for microalgae cell disruption [21]. It uses vibrations that break the cell structure mechanically, enhancing the extraction of lipids. Ultrasound pretreatment requires a lower energy demand, is effective, and is relatively economical [21]. In the present study, an ultrasound pretreatment for 5 min was applied. Our findings revealed that increasing the ultrasonication time from 5 (25.2%) to 15 min (25.6%) led to a slight increase in the lipid content obtained from *Nannochloropsis oculata* microalgae mixed with methanol: chloroform (1:2). In addition, a prolonged sonication time can significantly increase the temperature, protein denaturation, and liquid foaming [21]. Pohndorf et al. [12] investigated the influence of cell disruption (microwave, autoclaving, and milling) on lipid extraction from *Spirulina* sp. The results revealed that cell disruption by milling was more effective than microwave or autoclaving, probably due to the frictional forces that lead to the breaking of the particles, size reduction, and cell disruption [12].

The effect of methanol: chloroform ratio, biomass: solvent ratio, and extraction time on lipid extraction was analyzed. The polar and nonpolar solvent combination was chosen to extract the total lipids from *Spirulina* spp. cells. Generally, the polar solvent releases the lipid from their protein–lipid complexes, and further, the lipids dissolve quickly in the nonpolar solvent [22].

The average values of the lipid content (*SL*) and signal-to-noise (*S*/*N*) ratio for larger-the-better quality characteristics for each parameter at levels 1, 2, and 3 are presented in Table 1 for lipid extraction from *Spirulina* spp. The obtained *SL* values ranged from 4.93 to 8.05%.

Figure 1 was generated based on the *SL* and *S*/*N* values from the *L*_9_ orthogonal array to highlight the levels of the selected parameters with the highest influence on lipid extraction. A line connects the response for each level for *A*, *B*, and *C* parameters. The steeper the slope of the line is, the greater the magnitude of the main effect [23], as can be seen for the *B* parameter. The effect of the *A* and *C* parameters was almost negligible. Moreover, the effect of the *A* parameter on lipid extraction slowly increased when the ratio of polar and nonpolar solvents was equal, while those at the first and third levels decreased. In addition, the response curves revealed that the effect of the *C* parameter decreased with longer extraction times.

Table 1 indicates that the second level of parameter *A* (*A*_2_), the third level of parameter *B* (*B*_3_), and the first level of parameter *C* (*C*_1_) presented higher average values of *SL* and *S*/*N* ratio. Thus, the optimal levels of the parameters were methanol: chloroform ratio 1:1, biomass: solvent ratio 1:60, and extraction time of 30 min. The findings are in accordance with previous studies. Ryckebosch et al. [24] investigated different solvents and mixtures to extract microalgae lipids. Their results showed that chloroform–methanol 1:1 was the best solvent mixture for the extraction of total lipids. The extraction of lipids from *Spirulina* spp. biomass in a single-stage extraction with ethanol at room temperature was investigated by Chaiklahan et al. [25]. An increase in the solid-to-solvent ratio increased the yield of the extracted lipids. A solid: solvent ratio of 1:5 resulted in a lipid yield of 3.6%, while a solid: solvent ratio of 1:50 resulted in a lipid yield of 7.9% [25].

The following series was depicted based on the obtained delta values: *B* > *C* > *A* for both *SL* and *S*/*N* data. The *B* parameter strongly influenced *SL* value (the highest delta value). The smallest delta value was found for the *A* parameter.

Analysis of variance (ANOVA) was used to identify the significant parameters for the lipid extraction from *Spirulina* spp. by the sum of the squared deviations from the total mean of the *S*/*N* ratio [26] and the contribution of each parameter in the *L*_9_ [18]. In the ANOVA analysis, the total variation (sum of the squares) is equal to the sum of the squares of deviation of all the considered parameters [18]. The degrees of freedom (*DF*), the sum of squares (*SS*), mean square (*MS*), distribution of the ratio (*F*), and *p*-value [20] are shown in Table 2.

The *F*-ratio ensures the analysis hypothesis, and the *p*-value measures how much evidence we have against the null hypothesis (*H*_0_) [27]. A *p*-value <0.05 indicates significant parameters. The *p*-value (Table 2) for the *B* parameter was less than 0.05 (*p* < 0.02), which means strong evidence against the null hypothesis (*H*_0_). Consequently, the effect of this parameter on lipid extraction was statistically significant. Thus, the biomass: solvent ratio (*B*) was the most significant parameter for lipid extraction. The *p*-values for the *A* (methanol: chloroform ratio) and *C* parameters (extraction time) were 0.86 and 0.31, respectively. Thus, these parameters can be ignored as they were insignificant for lipid extraction (*p* > 0.05). 

The percent contribution of each parameter (*PC*), which reflects the relative portion of the total variation observed in an experiment attributed to each parameter [20], was calculated, and the results are given in Table 2.

The *PC* was calculated using the following equation:(1)$$PC=100\cdot \frac{\left(SS_{X}-V_{e}\right)\cdot v_{x}}{SS_{T}}$$
where *SS_X_* is the sums-of-squares for each parameter, *V_e_* is the mean-square-error, *v_x_* is the degree of freedom (*DF*) for the factor, and *SS_T_* is the total sum-of-squares [20].

Based on the *PC* value, the biomass: solvent ratio presented the highest contribution, 92.1% and 92.3%, to the *SL* and *S*/*N* data, respectively. Table 2 shows that the extraction time presented a contribution value of 5.3% and 5.0% for *SL* and *S*/*N* data, respectively. The M:C ratio presented the most negligible contribution below 1.0% for both *SL* and *S*/*N* data.

The mean at the optimum conditions (*μ*) was estimated after the identification of significant parameters from ANOVA analysis. *Μ* was calculated as follows: *μ* = *Ū* + (*B*_3_ − *Ū*)(2)
where *Ū* is the overall mean of the response and *B*_3_ is the average value of the response at the third level of parameter *B* [17,28].

The obtained *μ* value calculated using Equation (2) was found to be 8.1%. 

The estimation of *μ* is only based on the average of the obtained results from the experiments. As stated by Srivastava et al., from a statistical point of view, its value provides a 50% chance of the true average being greater than *μ* and a 50% chance of the true average being less than *μ* [17]. Thus, the confidence level range (*CI*) should be determined. As suggested by Taguchi, two confidence intervals should be determined, namely the confidence interval for a population (*CI_POP_*) and the confidence interval for a sample group (*CI_CE_*). *CI_POP_* and *CI_CE_* expressions are given as follows [17,28]:(3)$$CI_{POP}=\sqrt{\frac{F_{\alpha }\left(1,f_{e}\right)V_{e}}{n_{eff}}}$$
(4)$$CI_{CE}=\sqrt{F_{\alpha }\left(1,f_{e}\right)V_{e}\left[\frac{1}{n_{eff}}+\frac{1}{R}\right]}$$
(5)$$n_{eff}=\frac{N}{1+\left[\mathrm{TDOF}\right]}$$

Where *F_α_* (1, *f_e_*) represents the *F*-ratio at a confidence level of (1—α) against DOF = 1 and a DOF error of *f_e_*, *V_e_* is the error variance (from ANOVA), *R* represents the sample size for the confirmation experiment, *N* is the total number of results, and TDOF is the total DOF associated in the estimation of the mean [17,28].

The 95% confidence intervals for *CI_POP_* and *CI_CE_* are as follows:(6)$$CI_{POP}=\sqrt{\frac{F_{\alpha }\left(1,f_{e}\right)V_{e}}{n_{eff}}}=\pm \;1.92$$
(7)$$CI_{CE}=\sqrt{F_{\alpha }\left(1,f_{e}\right)V_{e}\left[\frac{1}{n_{eff}}+\frac{1}{R}\right]}=\pm \;2.22$$

The predicted optimum value for the lipid content *μ_L_* was calculated as follows:*μ_L_* = *Ū* + (*A*_2_ − *Ū*) + (*B*_3_ − *Ū*) + (*C_1_ − Ū*)(8)
where *μ_L_* is the predicted optimum value for lipid extraction; *A*_2_, *B*_3_, and *C*_1_ are the average lipid values at their optimum levels (Table 2) [17,28]. The obtained *μ_L_* value calculated using Equation (8) was 8.7%. 

### 2.2. Confirmation Experiments

Three experiments were carried out to validate the obtained results using the optimal process parameters identified (*A*_2_, *B*_3_, and *C*_1_)*_._* The results obtained are presented in Table 3. As shown in Table 3, the average value of the confirmation experiments was 8.7 ± 0.1% and was within a 95% of *CI_CE_*. In addition, three experiments were performed without ultrasound application. The average value of the experiments performed without ultrasound application was 7.1 ± 0.2%, lower than the average value of the confirmation experiment.

Process parameters, such as an osmotic NaCl concentration of 10–30%, solvent: biomass ratio of 5–15 *v*/*w*, and extraction times of 20–50 min, were investigated for the extraction of lipids from *Spirulina platensis* using a response surface methodology. The findings suggested that the applied osmotic shock method with ultrasound irradiation increased lipid yields to 6.65% using 11.9% NaCl, a solvent: biomass ratio of 12:1 *v*/*w*, and an extraction time of 22 min [29]. In addition, lipid yields ranging from 6.4 to 7.5% (DW) were reported for *Spirulina platensis*, *Spirulina maxima*, and *Spirulina* spp. food products using a mixture of dichloromethane/methanol (2:1) [30]. Kalsum et al. [31] reported a lipid extraction yield of 12.53% from *Spirulina platensis* after microwave-assisted extraction using a mixture of methanol: hexane (1:2) at 600 W for 40 min and a lipid extraction yield of 1.293% by the Soxhlet method after extraction using hexane for 40 h. The lipid extraction yield from microalgae depends on the solvent type and selectivity [31]. Previous work has evaluated the effectiveness of green solvents, such as supercritical CO_2_, high-pressure ethanol, and supercritical CO_2_ with ethanol as a co-solvent, compared to Soxhlet, Bligh and Dyer, and ultrasound-assisted extraction for the lipid extraction from *Spirulina platensis*. The highest lipid content was obtained using the Bligh and Dyer method (11.6%), followed by high-pressure ethanol (11.4%) [32].

### 2.3. Fatty Acid Composition (as Fatty Acid Methyl Esters, FAME)

The most predominant saturated (SFAs), monounsaturated (MUFAs), and polyunsaturated fatty acids (PUFAs) identified in *Spirulina* spp. lipid extract are presented in Table 4. Generally, no differences in the fatty acid composition were observed between the samples obtained after extraction with and without ultrasound application.

Palmitic acid (C16:0) was found to be the predominant SFA in *Spirulina* spp. (44.5%). Similar results were reported for autotrophically cultivated *Spirulina platensis* and *Spirulina platensis* products [30]. Palmitoleic acid (C16:1), a MUFA, was found in a small amount in *Spirulina* spp. (3.3%). The content of linoleic acid (14.9%) in *Spirulina* spp. was approximatively the same as reported for *Spirulina platensis* (13.6%) [30]. Linoleic and gamma-linolenic acids were found as the most abundant PUFAs. The content of gamma-linolenic acid (13.4%) was almost similar to that reported for *Spirulina platensis* (15.2%) [29].

## 3. Materials and Methods

### 3.1. Reagents and Solutions

The *Spirulina* spp. biomass was purchased from the Romanian market. The chemicals, such as methanol (CH_3_OH) and chloroform (CHCl_3_) of analytical grade, were purchased from Merck (Darmstadt, Germany). 

### 3.2. Taguchi Method of Experimental Design

The Taguchi methodology consists of four phases with various steps: planning, conducting the experiments, analyzing experimental results, and validating the obtained results [17].

#### 3.2.1. Design of Experiment (Phase 1)

The first step in Phase 1 was identifying the parameters that need to be optimized and affect the lipid extraction process. Three parameters with three levels each were considered to identify the optimum conditions for lipid extraction from *Spirulina* spp. The selected parameters and their levels are presented in Table 5.

After selecting the parameters and their levels, the next step was to design the matrix of experiments and select the data analysis procedure [17]. The parameters and their levels presented in Table 5 were introduced in the Minitab 15 software. Using the Design of Experiment (DOE) function of the software, the *L*_9_ orthogonal array was generated based on the combination of the considered parameters and their levels. The *L*_9_ orthogonal array is given in Table 6. 

#### 3.2.2. Lipid Extraction (Phase 2)

The schematic representation of lipid extraction from *Spirulina* spp. is presented in Figure 2. The experiments were carried out according to the *L*_9_ orthogonal array shown in Table 6 by contacting an amount of 1 g of *Spirulina* spp. with various solvent mixtures (1:10, 1:30, and 1:60) such as: methanol: chloroform (MC 2:1, *v*/*v*), methanol: chloroform (MC 1:1, *v*/*v*), and methanol: chloroform (MC 1:2, *v*/*v*). The samples, placed in conical flasks, were subjected to ultrasonication for 5 min at a frequency of 35 kHz using an ultrasonic batch (Sonorex RK 512 H) and further stirred at 500 rpm using a magnetic stirrer at room temperature for 30–120 min. Preliminary tests showed that an increase in the temperature up to 60 °C led to a slight decrease in the lipid content. After extraction, the *Spirulina* spp. biomass was separated from the liquid extract by centrifugation at 4000 rpm for 10 min, followed by filtration. H_2_O (20% of the total volume) was added to the liquid extract for phase separation and extract purification. The chloroform fraction was separated and evaporated using a rotary evaporator Laborota 4010 (Heidolph, Schwabach, Germany), dried in an oven at 60 °C, and finally weighed. The lipid content was determined gravimetrically using an analytical balance (Partner, XA 220, Radwag, Radom, Poland) accurate to 0.00001 g. The lipid content (*SL*) was calculated using the following equation:(9)$$\left(SL\right)\left(\%\right)=\frac{m_{1}}{m_{0}}\cdot 100$$
where *m*_l_ is the mass of extracted lipids (g) and *m*_0_ is the mass of dried microalgae powder (g) [33,34,35].

The experimental results expressed in terms of lipid content and dry weight (DW), based on Taguchi’s orthogonal array, are presented in Table 6.

#### 3.2.3. Analysis of Experimental Data and Prediction of Performance (Phase 3)

Taguchi created the *S*/*N* ratio, which combines the mean level of the quality characteristics and the variance into a single metric [17].

The experimental data were processed using larger-the-better quality characteristics to determine the optimum lipid extraction conditions and estimate the lipid content at the optimum conditions. The equation for calculating the *S*/*N* ratio for larger-the-better quality characteristics is given as follows:(10)$$S/N=-10\;log\left[\left(\sum \frac{1}{Y^{2}}\right)/n\right]$$
where *Y* is the value of the response and *n* is the total number of repetitions in a trial [20].

Minitab software was used to calculate the equivalent *S*/*N* ratio of the experiments based on the obtained values using the larger-the-better equation to maximize the response [26]. The obtained results are presented in Table 6.

The response curves for the individual effects, ANOVA analysis, and *S*/*N* data were used for the data analysis. The *PC* for each parameter was calculated. In addition, the confidence intervals, namely *CI_POP_* and *CI_CE_*, were determined.

#### 3.2.4. Confirmation Experiment (Phase 4)

The Taguchi method’s final step is verifying the conclusions obtained from the experiments. Thus, three experiments were carried out to validate the obtained results using the identified optimum conditions. The average value of the results obtained from confirmation experiments was compared with the predicted average value.

### 3.3. Preparation of Fatty Acid Methyl Esters (FAMEs) and GC Analysis

The lipids were converted to FAMEs by transesterification. Briefly, 4 mL of isooctane was added to the lipid extract (0.06 g) in a conical-bottom glass centrifuge tube and then treated with 0.2 mL of potassium hydroxide solution in methanol (2 mol/L). After 30 s of vigorous stirring, 1 g of NaHSO_4_·H_2_O was added to prevent methyl esters saponification and to neutralize the excess alkali. The FAMEs composition was analyzed using a gas chromatograph with a flame ionization detector (GC-FID) (Agilent Technologies, 6890N GC, Wilmington, DE, USA). The samples were injected into a DB-WAX capillary column (30 m × 0.25 mm × 0.25 µm) and eluted with helium (purity ≥ 99.999%) at a constant flow rate of 1.53 mL/min and a pressure of 70 kPa. The injection volume was 1 µL at a split ratio of 1:20. The temperature of the oven was set as follows: 60 °C for 1 min, 60 to 200 °C (rate 10 °C/min, 2 min), and from 200 to 220 °C (5 °C/min, 20 min). The injector and detector temperatures were set to 250 °C. FAME components were identified by comparing their retention times with those of the standard mixture (Supelco 37 FAME Mix, Sigma-Aldrich, Saint Louis, USA).

## 4. Conclusions

In the present study, the Taguchi method of experimental design was used to extract lipids from cyanobacteria *Spirulina* spp. by ultrasound application and mechanical stirring. Parameters, such as methanol: chloroform (M:C) ratio, biomass: solvent ratio, and extraction time at three levels, were optimized with larger-the-better quality characteristics. The optimum conditions were determined to be: M:C ratio of 1:1, biomass: solvent ratio of 1:60, and extraction time of 30 min. Based on the percent contribution of each parameter value, the biomass: solvent ratio had the highest contribution (92.3%) for the lipid extraction process, while extraction time had a contribution value of 5.0%. Methanol: chloroform ratio had the lowest contribution (below 1.0%). The value of lipid extraction obtained through the confirmation experiments was within a 95% confidence interval. The obtained value of confirmation experiments for lipid extraction (8.7 ± 0.1%) was in good agreement with the predicted value (8.7%) and higher than the average value of the experiments performed without ultrasound application (7.1 ± 0.2%). Thus, the results of the experiments showed that the Taguchi method effectively predicted the response. The obtained results can be helpful for the development of efficient methods for the production of the third generation of biofuels from feedstocks with low lipid content.

## Figures and Tables

**Figure 1 molecules-27-06794-f001:**
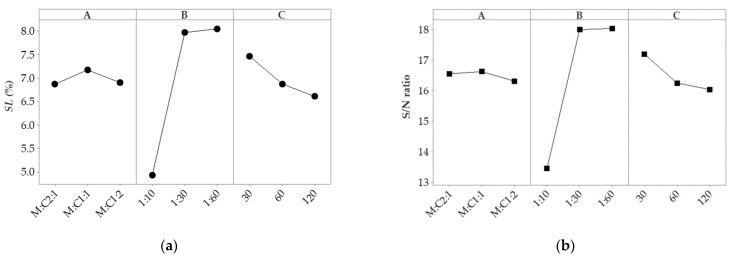
Effect of process parameters on lipid extraction from *Spirulina* spp.: (**a**) *SL*; (**b**) *S*/*N* ratio.

**Figure 2 molecules-27-06794-f002:**
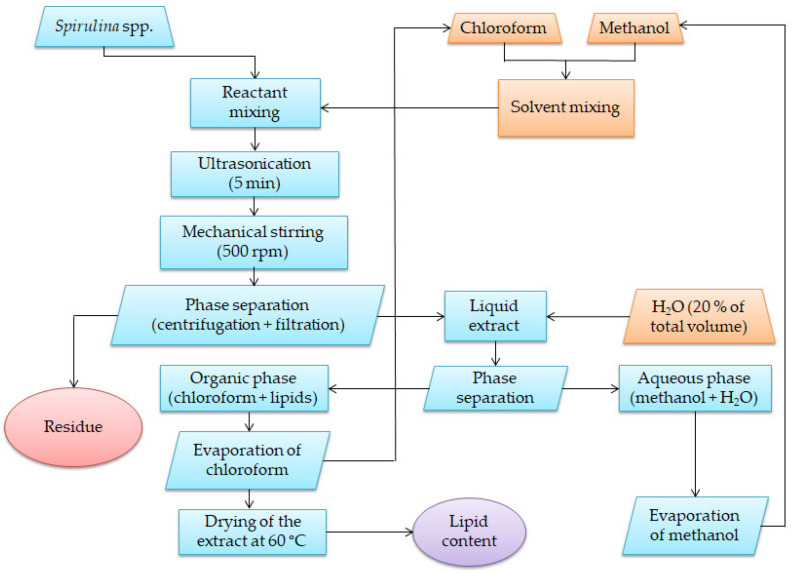
Schematic representation of lipid extraction from *Spirulina* spp.

**Table 1 molecules-27-06794-t001:** The average *SL* and *S*/*N* ratio values.

	*SL*			*S*/*N*		
Level	*A*	*B*	*C*	*A*	*B*	*C*
1	6.87	4.93	7.46	16.56	13.45	17.20
2	7.17	7.97	6.87	16.62	18.00	16.25
3	6.91	8.05	6.61	16.31	18.04	16.04
Delta	0.30	3.12	0.85	0.31	4.59	1.16
Rank	3	1	2	3	1	2

**Table 2 molecules-27-06794-t002:** ANOVA for SL and *S*/*N* ratio for lipid extraction from *Spirulina* spp.

	*SL*							*S*/*N*						
Parameter	*DF*	*Seq SS*	*Adj SS*	*Adj MS*	*F*	*p*	*PC* (%)	*DF*	*Seq SS*	*Adj SS*	*Adj MS*	*F*	*p*	*PC* (%)
A	2	0.17	0.17	0.08	0.41	0.71	0.9	2	0.16	0.16	0.08	0.16	0.86	0.4
B	2	19.00	19.00	9.48	47.11	0.02	92.1	2	41.79	41.79	20.90	41.38	0.02	92.3
C	2	1.15	1.15	0.60	2.84	0.26	5.3	2	2.29	2.29	1.15	2.27	0.31	5.0
RE *	2	0.40	0.40	0.20				2	1.01	1.01	0.51			
Total	8	20.68						8	45.25					

* RE-residual error.

**Table 3 molecules-27-06794-t003:** Predicted optimal lipid content and the results of confirmation experiment.

Microalgae	Predicted Optimal Value (%)	Confidence Intervals (95%)	Confirmation Experiment (%)
*Spirulina* spp.	8.7	*CI_POP_*: 6.8 < *µ_L_* < 10.6	8.7 ± 0.1
		*CI_CE_*: 6.5 < *µ_L_* < 10.9	

**Table 4 molecules-27-06794-t004:** Fatty acid composition of lipid extract from *Spirulina* spp. with and without ultrasound application.

Fatty Acids	Class of Fatty Acids	Ultrasound %	Without Ultrasound (%)
Palmitic acid, C16:0	SFA	44.5	42.9
Palmitoleic acid, C16:1	MUFA	3.3	3.2
Linoleic acid, C18:2(cis + trans) (n6)	PUFA	14.9	14.4
Gamma-linolenic acid, C18:3(n6)	PUFA	13.4	15.4

**Table 5 molecules-27-06794-t005:** Process parameters and their levels for lipid extraction.

Symbol	Parameters	Level 1	Level 2	Level 3
A	methanol:chloroform ratio (*v*/*v*)	M:C 2:1	M:C 1:1	M:C 1:2
B	biomass: solvent ratio (*w*/*v*)	1:10	1:30	1:60
C	extraction time (min)	30	60	120

**Table 6 molecules-27-06794-t006:** Design of experiments using Taguchi orthogonal array and experimental SL and calculated S/N values.

Experiment	*A*	*B*	*C*	*SL* *	*S*/*N*
1	M:C 2:1	1:10	30	5.6 ± 0.9	14.7
2	M:C 2:1	1:30	60	7.7 ± 0.4	17.7
3	M:C 2:1	1:60	120	7.3 ± 0.1	17.3
4	M:C 1:1	1:10	60	4.8 ± 1.2	13.0
5	M:C 1:1	1:30	120	8.1 ± 0.1	18.1
6	M:C 1:1	1:60	30	8.7 ± 0.1	18.8
7	M:C 1:2	1:10	120	4.4 ± 0.7	12.7
8	M:C 1:2	1:30	30	8.1 ± 0.5	18.2
9	M:C 1:2	1:60	60	8.1 ± 1.0	18.1
Mean				7.0	

* Mean values ± standard deviation (of 3 repetitions).

## Data Availability

The data presented in this study are available on request from the corresponding author.

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
