# Peer review of "Optimization of Lipid Extraction from Spirulina spp. by Ultrasound Application and Mechanical Stirring Using the Taguchi Method of Experimental Design"

_molecules, 2022, doi:10.3390/molecules27206794_

Round 1

Reviewer 1 Report

Manuscript title: Optimization of ultrasound-assisted extraction of lipids from Spirulina spp. using Taguchi method of experimental design

Emilia Neag*, Zamfira Stupar, Cerasel Varaticeanu, Marin Senila* and Cecilia Roman

The manuscript is a good attempt to gather information about a new method for lipids extraction from Spirulina spp.

Materials and Methods are written well and detailed. However, minor revision is required before publication. Please find the comments below

1. Page 1, lines 37-42: the name of all used solvents is not required to be detailed, it is better to state some most important ones.

2. Page 6, Table 5: the line numbers are included in the table, which must be edited.

3. Page 6, line 209: the”ultrasonic batch at 35 kHz”. It is ultrasonic bath

4. Discussion is weak and it should be better written, by this way a comparison of the other extraction methods including supercritical fluid extractions, which the Taguchi design is used generally, or other methods such as cold press, microextraction, microwave-assisted methods etc.

5. Some images or schematics for the extraction process will advance the manuscript.

6. The conclusion is written so brief, it should be improved and some future prospects must be included

Author Response

Esteemed Reviewer,

Thank you for your very careful review of our paper, and for the comments, corrections and suggestions. Your comments have helped us to carry out a major revision of the paper in order to improve the presentation of the obtained results. Please find below the list with the modifications as requested. Changes were highlighted using “Track Changes” function. We hope that the revised manuscript would meet the Reviewer’s expectations.

Point 1: Manuscript title: Optimization of ultrasound-assisted extraction of lipids from Spirulina spp. using Taguchi method of experimental design Emilia Neag*, Zamfira Stupar, Cerasel Varaticeanu, Marin Senila* and Cecilia Roman. The manuscript is a good attempt to gather information about a new method for lipids extraction from Spirulina spp. Materials and Methods are written well and detailed. However, minor revision is required before publication.

Please find the comments below.

Page 1, lines 37-42: the name of all used solvents is not required to be detailed, it is better to state some most important ones.

Answer: Done.

Point 2: Page 6, Table 5: the line numbers are included in the table, which must be edited.

Answer: Done.

Point 3:  Page 6, line 209: the”…ultrasonic batch at 35 kHz”. It is ultrasonic bath

Answer: The sentence was rephrased.

Point 4: Discussion is weak and it should be better written, by this way a comparison of the other extraction methods including supercritical fluid extractions, which the Taguchi design is used generally, or other methods such as cold press, microextraction, microwave-assisted methods etc.

Answer: The discussion section was improved.

Pohndorf et al., [12] investigated the influence of cell disruption (microwave, autoclaving, and milling) on lipid extraction from Spirulina sp. The results revealed that cell disruption by milling was more effective than microwave or autoclaving, probably due to the frictional forces that lead to the breaking of the particles, size reduction, and cell disruption [12].

The findings are in accordance with previous studies. Ryckebosch et al., investigated different solvents and mixtures to extract microalgae lipids. Their results showed that chloroform–methanol 1:1 was the best solvent mixture for the extraction of total lipids [24]. The extraction of lipids from Spirulina spp. biomass in a single-stage extraction with ethanol at room temperature was investigated by Chaiklahan et al. [25]. An increase in the solid to solvent ratio increased the yield of the extracted lipids. A solid: solvent ratio of 1:5 resulted in a lipid yield of 3.6 %, while a solid: solvent ratio of 1:50 resulted in a lipid yield of 7.9 % [25].

Process parameters, such as osmotic NaCl concentration of 10 – 30 %, solvent: biomass ratio of 5 – 15 v/w, and extraction times of 20 – 50 min were investigated for the extraction of lipids from Spirulina platensis using response surface methodology. The findings suggested that the applied osmotic shock method with ultrasound irradiation increased lipid yields to 6.65 % using 11.9 % NaCl, a solvent: biomass ratio of 12:1 v/w, and an extraction time of 22 min [29]. Also, lipids yields ranging from 6.4 to 7.5 % (DW) were reported for Spirulina platensis, Spirulina maxima, and Spirulina spp. food products using a mixture of dichloromethane/methanol (2:1) [30]. Kalsum et al., [31] reported a lipid extraction yield of 12.53 % from Spirulina platensis after microwave-assisted extraction using a mixture of methanol: hexane (1:2) at 600 W for 40 min and a lipid extraction yield of 1.293 % by Soxhlet method after extraction using hexane for 40 h. The lipid extraction yield from microalgae depends on the solvent type and selectivity [31]. Previous work has evaluated the effectiveness of green solvents, such as supercritical CO2, high-pressure ethanol, and supercritical CO2 with ethanol as a co-solvent, compared to Soxhlet, Bligh & Dyer, and ultrasound-assisted extraction for the lipid extraction from Spirulina platensis. The highest lipid content was obtained using Bligh & Dyer method (11.6 %), followed by high-pressure ethanol (11.4 %) [32].

The following references were added:

31.Kalsum, U.; Kusuma, H.S.; Roesyadi, A.; Mahfud, M. Lipid Extraction from Spirulina platensis using microwave for biodiesel production, Korean Chem. Eng. Res. 2019, 57(2), 301–304.

22.Bobde, K.; Momin, H.; Bhattacharjee, A.; Aikat, K. Energy assessment and enhancement of the lipid yield of indigenous Chlorella sp. KA-24NITD using Taguchi approach. Renew. Energ. 2019, 131, 1226–1235.

23.Anıl, I.; Öztürk, N.; Alagha, O.; Ergenekon, P.; Optimization of solid-phase microextraction using Taguchi design to quantify trace level polycyclic aromatic hydrocarbons in water. J. Sep. Sci. 2012, 35, 3561–3568.

24.Ryckebosch, E.; Muylaert, K.; Foubert, I.; Optimization of an analytical procedure for extraction of lipids from microalgae. J. Am. Oil Chem. Soc. 2012, 89(2) 189–198.

21.Rokicka, M., ZieliÅ„ski, M., Dudek, M.; DÄ™bowski, M. Effects of ultrasonic and microwave pretreatment on lipid extraction of microalgae and methane production from the residual extracted biomass. Bioenerg. Res. 2021, 14, 752–760.

32.Ríos Pinto, L.F.; Ferreira, G.F.; Beatriz, F.P.; Cabral, F.A.; Filho, R.M. Lipid and phycocyanin extractions from Spirulina and economic assessment. J. Supercrit. Fluids. 2022, 184, 105567.

29.Hadiyanto, H., Adetya, N.P. Response surface optimization of lipid and protein extractions from Spirulina platensis using ultrasound assisted osmotic shock method. Food Sci. Biotechnol. 2018, 27, 1361–1368.

30.Ambrozova, J.V.; Misurcova, L.; Vicha, R.; Machu, L.; Samek, D.; Baron, M.; Mlcek, J.; Sochor, J.; Jurikova, T. Influence of extractive solvents on lipid and fatty acids content of edible freshwater algal and seaweed products, the green microalga Chlorella kessleri and the cyanobacterium Spirulina platensis. Molecules 2014, 19, 2344-2360.

25.Chaiklahan, R.; Chirasuwan, N.; Loha, V.; Bunnag, B.; Lipid and fatty acids extraction from the cyanobacterium Spirulina. Sci. Asia. 2008, 34, 299–305.

Point 5: Some images or schematics for the extraction process will advance the manuscript.

Answer: The schematic representation of lipid extraction from Spirulina spp. is presented in Figure 2 in 3.2.2 Lipid extraction (Phase 2).

Point 6:  The conclusion is written so brief, it should be improved and some future prospects must be included

Answer: The conclusion section was improved. In the present study, the Taguchi method of experimental design was used to extract lipids from cyanobacteria Spirulina spp. by ultrasound application and mechanical stirring. Parameters, such as methanol: chloroform (M:C) ratio, biomass: solvent ratio, and extraction time at three levels, were optimized with larger-the-better quality characteristics. The optimum conditions were determined to be: M:C ratio of 1:1, biomass: solvent ratio of 1:60 and extraction time of 30 min. Based on the percent contribution of each parameter value, the biomass: solvent ratio had the highest contribution (92.3 %), for the lipid extraction process, while extraction time had a contribution value of 5.0 %. Methanol: chloroform ratio had the lower contribution (below 1.0 %). The value of lipid extraction obtained through the confirmation experiments was within 95 % confidence intervals. The obtained value of confirmation experiments for lipid extraction (8.7 ± 0.1 %) was in good agreement with the predicted value (8.7 %) and higher than the average value of the experiments performed without ultrasounds application (7.1 ± 0.2 %). Thus, the results of the experiments showed that the Taguchi method effectively predicted the response. The obtained results can be helpful for the development of efficient methods for the production of the third generation of biofuels from feedstocks with low lipid content.

Reviewer 2 Report

The authors of the presented manuscript attempted to optimise the lipid extraction process from Spirulina spp. They checked relevant parameters, such as methanol:chloroform ratio, biomass : solvent ratio, extraction time at three levels were optimised with "larger-the-better" quality characteristics using 9 sets of experiments. The studies were correctly planned and conducted. The authors obtained reliable results that were well analysed. The work can be published in its present form. I have two minor editorial comments: 1) page 5 line 156 please remove "M3"; page 5 line 167 please remove the full stop at the end of subtitle 3.2.1.

Author Response

Esteemed Reviewer,

Thank you for your very careful review of our paper, and for the comments, corrections and suggestions.

Point 1: The authors of the presented manuscript attempted to optimise the lipid extraction process from Spirulina spp. They checked relevant parameters, such as methanol:chloroform ratio, biomass: solvent ratio, extraction time at three levels were optimised with "larger-the-better" quality characteristics using 9 sets of experiments. The studies were correctly planned and conducted. The authors obtained reliable results that were well analysed. The work can be published in its present form.

I have two minor editorial comments: 1) page 5 line 156 please remove "M3";

Answer: Done

Point 2: page 5 line 167 please remove the full stop at the end of subtitle 3.2.1.

Answer:  Done

Reviewer 3 Report

The manuscript is well-written. I have some comments/observations:

The extraction can be further maximized:

a) Did you treat the biomass prior to the extractions? Some authors demonstrated that this is an important issue for the extraction of lipids from Spirulina. Some methods might extreamly impact on cell disruption. In this way, the time of extraction can be decreased. 

b) why didn't you considered to include also the temperature? The temperature has been demonstrated to influence the extraction of lipid from Spirulina. 

c) In the introduction you reported that ultrasound treatment is convenient. Why didn't you consider this as a parameter of DOE? Also ultrasound might decrease the time. A total of 5 min before the magnetic stirring might not be enough to influence the extraction in terms of yield.

d) Same as above, why did you consider the pressure?

- Why didn't you characterize the extracted lipids in terms of composition? The variation of the solvent can condition the percentage of neutral and polar lipids. 

- Provide more information regarding the ANOVA. Did you perform post-hoc test?

- Provide the standard deviations not only the mean. 

- Explain better the meaning and the interpretation of S/N values and the performance characteristics “lower-the-better”, “larger-the-better” and “nominal-the-better”. A reader that is not expert on Taguchi’s methodology might not understand everything. 

The discussion of the results is missing. Talk about the advantages of your method compared to others. 

Author Response

Esteemed Reviewer,

Thank you for your very careful review of our paper and for the constructive comments concerning our manuscript. A major revision of the paper has been carried out to take all of them into account. We answer your questions and comments in details in the following texts. Changes were highlighted using “Track Changes” function. We hope that the revised manuscript would meet the Reviewer’s expectations.

Point 1: The manuscript is well-written. I have some comments/observations: The extraction can be further maximized:

a) Did you treat the biomass prior to the extractions? Some authors demonstrated that this is an important issue for the extraction of lipids from Spirulina. Some methods might extreamly impact on cell disruption. In this way, the time of extraction can be decreased. 

Answer: According to the literature, ultrasound is one of the most promising pretreatment methods for microalgal cell disruption. It uses vibrations that break the cell structure mechanically, enhancing the extraction of lipids. Ultrasound pretreatment requires a lower energy demand, is effective, and is relatively economical [21]. In the present study, an ultrasound pretreatment for 5 min was applied. A prolonged sonication time can lead significantly increase the temperature, protein denaturation, and liquid foaming [21]. Pohndorf et al., [12] investigated the influence of cell disruption (microwave, autoclaving, and milling) on lipid extraction from Spirulina sp. The results revealed that cell disruption by milling was more effective than microwave or autoclaving, probably due to the frictional forces that lead to the breaking of the particles, size reduction, and cell disruption [12].

Point 2: b) why didn't you considered to include also the temperature? The temperature has been demonstrated to influence the extraction of lipid from Spirulina. 

Answer: Our preliminary studies showed that an increase in the temperature from 293 to 333 K led to a decrease in the lipid yield for Spirulina spp. and Nannochloropsis oculata, respectively. The experimental conditions were as follows: 2 g of microalgal biomass, stirred speed of 500 rpm, solvent to solid ratio 20:1, chloroform: methanol 2:1.

Point 3: c) In the introduction you reported that ultrasound treatment is convenient. Why didn't you consider this as a parameter of DOE? Also ultrasound might decrease the time. A total of 5 min before the magnetic stirring might not be enough to influence the extraction in terms of yield.

Answer: We didn’t consider ultrasound treatment as a parameter for DOE due to the fact a prolonged sonication time can lead significantly increase the temperature, protein denaturation, and liquid foaming [21]. Also, we conducted the following experiments: Nanochlorposis ocultata microalgae was contacted with methanol: chloroform (1:2) for a ultrasonication time of 5 min, 10 min and 15 min and we obtained a lipid content of 25.2%, 25.4% and 25.6 %, respectively. A small variation was observed in terms of lipid content after increasing the ultrasonication time. Thus, based on the obtained results we choose an ultrasound pretreatment for 5 min to reduce the energy costs.

Point 4: d) Same as above, why did you consider the pressure?

Answer: The ultrasonic batch (Sonorex RK 512 H) generates waves only at a frequency pressure of 35 kHz.

- Why didn't you characterize the extracted lipids in terms of composition? The variation of the solvent can condition the percentage of neutral and polar lipids. 

Answer: The fatty acid composition (as fatty acid methyl esters, FAME) is given in section 2.3. The most predominant saturated (SFAs), monounsaturated (MUFAs) and polyunsaturated fatty acids (PUFAs) identified in Spirulina spp. lipid extract are presented in Table 4. Generally, no differences in the fatty acid composition were observed between the samples obtained after extraction with and without ultrasound application. Palmitic acid (C16:0) was found to be the predominant SFA in Spirulina spp. (44.5%). Similar results were reported for autotrophically cultivated Spirulina platensis and Spirulina platensis products [30]. Palmitoleic acid (C16:1), a MUFA, was found in a small amount in Spirulina spp. (3.3 %). The content of linoleic acid (14.9 %) in Spirulina spp. was approximatively the same as reported for Spirulina platensis (13.6 %) [30]. Linoleic and gamma-linolenic acids were found as the most abundant PUFAs. The content of gamma-linolenic acid (13.4 %) was almost similar to that reported for Spirulina platensis (15.2%) [29].

Provide more information regarding the ANOVA. Did you perform post-hoc test?

Answer: We didn’t performed post-hoc test. The confirmation experiments indicated the lipid content values in good agreement with the predicted value, within a 95 % confidence interval, proving the Taguchi method’s effectiveness in optimizing the process parameters for lipid extraction.

More information regarding ANOVA analysis were given. Analysis of variance (ANOVA) was used to identify the significant parameters for the lipid extraction from Spirulina spp. by the sum of the squared deviations from the total mean of the S/N ratio [26] and the contribution of each parameter in the L9 [18]. In the ANOVA analysis, the total variation (sum of the squares) is equal to the sum of the squares of deviation of all the considered parameters [18]. A p-value < 0.05 indicates the significant parameters. The p-values (Table 2) for the B parameter is less than 0.05 (p < 0.02), which means strong evidence against the null hypothesis (H0). Consequently, the effect of this parameter on lipid extraction is statistically significant. Thus, the biomass: solvent ratio (B) is the most significant parameter for lipid extraction. The p-value for A (methanol: chloroform ratio) and C parameters (extraction time) were 0.86 and 0.31, respectively. Thus, these parameters can be ignored as they are insignificant for lipid extraction (p > 0.05). The percent contribution of each parameter (PC), which reflects the relative portion of the total variation observed in an experiment attributed to each parameter [20].

Provide the standard deviations not only the mean. 

Answer: Done

Explain better the meaning and the interpretation of S/N values and the performance characteristics “lower-the-better”, “larger-the-better” and “nominal-the-better”. A reader that is not expert on Taguchi’s methodology might not understand everything. 

Answer: According to the purpose of the experiments, the lower-the-better equation is used to minimize the system response (quality decreases as the system response increases), and the larger-the-better equation is used to maximize the system response (quality increases as the system response increases). In contrast, the nominal-the-better equation is used to achieve a particular target value [20].

The discussion of the results is missing. Talk about the advantages of your method compared to others. 

Answer: The extraction of lipids from Spirulina spp. after ultrasonic and mechanical stirring was evaluated. According to the literature, ultrasound is one of the most promising pretreatment methods for microalgal cell disruption. It uses vibrations that break the cell structure mechanically, enhancing the extraction of lipids. Ultrasound pretreatment requires a lower energy demand, is effective, and is relatively economical [21]. In the present study, an ultrasound pretreatment for 5 min was applied. A prolonged sonication time can lead significantly increase the temperature, protein denaturation, and liquid foaming [21]. Pohndorf et al., [12] investigated the influence of cell disruption (microwave, autoclaving, and milling) on lipid extraction from Spirulina sp. The results revealed that cell disruption by milling was more effective than microwave or autoclaving, probably due to the frictional forces that lead to the breaking of the particles, size reduction, and cell disruption [12].

The effect of methanol: chloroform ratio, biomass: solvent ratio, and extraction time on lipid extraction was analyzed. The polar and non-polar solvent combination was chosen to extract the total lipids from Spirulina spp. cell. Generally, the polar sol-vent releases the lipid from their protein-lipid complexes, and further, the lipids dis-solve quickly in the non-polar solvent [22].

Figure 1 was generated based on the SL and S/N values from the L9 orthogonal array to highlight the levels of the selected parameters with the highest influence on lipid ex-traction. A line connects the response for each level for A, B, and C parameters. The steeper the slope of the line is, the greater the magnitude of the main effect [23], as it can be seen for the B parameter. The effect of the A and C parameters is almost negligible. Moreover, the effect of the A parameter on lipid extraction slowly increased when the ratio of polar and non-polar solvents was equal, while at the first and third levels decreased. Also, the response curves revealed that the effect of the C parameter de-creased with longer extraction times.

The findings are in accordance with previous studies. Ryckebosch et al., investigated different solvents and mixtures to extract microalgae lipids. Their results showed that chloroform–methanol 1:1 was the best solvent mixture for the extraction of total lipids [24]. The extraction of lipids from Spirulina spp. biomass in a single-stage extraction with ethanol at room temperature was investigated by Chaiklahan et al. [25]. An increase in the solid to solvent ratio increased the yield of the extracted lipids. A solid: solvent ratio of 1:5 resulted in a lipid yield of 3.6 %, while a solid: solvent ratio of 1:50 resulted in a lipid yield of 7.9 % [25].

 Process parameters, such as osmotic NaCl concentration of 10 – 30 %, solvent: biomass ratio of 5 – 15 v/w, and extraction times of 20 – 50 min were investigated for the extraction of lipids from Spirulina platensis using response surface methodology. The findings suggested that the applied osmotic shock method with ultrasound irradiation increased lipid yields to 6.65 % using 11.9 % NaCl, a solvent: biomass ratio of 12:1 v/w, and an extraction time of 22 min [29]. Also, lipids yields ranging from 6.4 to 7.5 % (DW) were reported for Spirulina platensis, Spirulina maxima, and Spirulina spp. food products using a mixture of dichloromethane/methanol (2:1) [30]. Kalsum et al., [31] reported a lipid extraction yield of 12.53 % from Spirulina platensis after microwave-assisted extraction using a mixture of methanol: hexane (1:2) at 600 W for 40 min and a lipid extraction yield of 1.293 % by Soxhlet method after extraction using hexane for 40 h. The lipid extraction yield from microalgae depends on the solvent type and selectivity [31]. Previous work has evaluated the effectiveness of green solvents, such as supercritical CO2, high-pressure ethanol, and supercritical CO2 with ethanol as a co-solvent, compared to Soxhlet, Bligh & Dyer, and ultrasound-assisted extraction for the lipid extraction from Spirulina platensis. The highest lipid content was obtained using Bligh & Dyer method (11.6 %), followed by high-pressure ethanol (11.4 %) [32].

Reviewer 4 Report

Under the optimized extraction conditions, an experimental control without applying ultrasound would have been useful.

It would have been interesting to quantify what percentage could increase lipid recovery in Spirulina spp. with and without ultrasound-assisted extraction.

Some observations have been noted directly in the pdf of the manuscript.

Author Response

Esteemed Reviewer,

Thank you for your very careful review of our paper, and for the comments, corrections and suggestions. Your comments have helped us to carry out a major revision of the paper in order to improve the presentation of the obtained results.

Point 1: Under the optimized extraction conditions, an experimental control without applying ultrasound would have been useful.

Answer: Three experiments were performed without ultrasound application. The average value of the experiments performed without ultrasound application was 7.1 ± 0.2 %. The obtained results are presented in 2.2 section.

Point 2: It would have been interesting to quantify what percentage could increase lipid recovery in Spirulina spp. with and without ultrasound-assisted extraction.

Answer: Three experiments were performed without ultrasounds application. The average value of the experiments performed without ultrasound application was 7.1 ± 0.2 %. The obtained results are presented in 2.2 section.

Point 3: Some observations have been noted directly in the pdf of the manuscript.

Answer:  All the observations were addressed. Please see the attached document.

Reviewer 5 Report

In the presented manuscript, authors used experimental design, based on so called Taguchi methodology to assess the most favorable conditions for the ultrasound-assisted extraction of lipids from microalgae Spirulina spp. The methodology used is presented with enough details but the results and discussion part lack some proper discussion. With the emphasis on the “optimization” of the process, I must address the amount of the solvent which is obtained as “optimal”. Biomass:solvent ratio of 1:60 is considerably larger than the ratio 1:30, but SL and SN percentages, are only slightly higher. Is this difference enough to justify such increase in solvent usage in practical exepriments (having green chemistry in mind and the use of ultrasound-assisted extraction due to its lower consumption of solvents, as also stated in the Introduction), or, are the authors strictly focused on mathematical numbers obtained from the calculations based on “larger-the-better” characteristics? Some of these issues should be discussed.

Is Biomass:solvent ratio given as v/v or m/v?

In Conformation experiment, SD value should be provided for the average value of three experiments.

Author Response

Esteemed Reviewer,

Thank you for the constructive comments and suggestions made to our manuscript. Please find below the list with the modifications as requested.

Point 1: In the presented manuscript, authors used experimental design, based on so called Taguchi methodology to assess the most favorable conditions for the ultrasound-assisted extraction of lipids from microalgae Spirulina spp. The methodology used is presented with enough details but the results and discussion part lack some proper discussion. With the emphasis on the “optimization” of the process, I must address the amount of the solvent which is obtained as “optimal”. Biomass:solvent ratio of 1:60 is considerably larger than the ratio 1:30, but SL and SN percentages, are only slightly higher. Is this difference enough to justify such increase in solvent usage in practical experiments (having green chemistry in mind and the use of ultrasound-assisted extraction due to its lower consumption of solvents, as also stated in the Introduction), or, are the authors strictly focused on mathematical numbers obtained from the calculations based on “larger-the-better” characteristics? Some of these issues should be discussed.

Answer: Indeed, a smaller solvent-to-solid ratio could lead to a decreased in microalgal lipid yield, but a higher solvent-to-solid ratio could be not feasible from economic point of view. The extraction of lipids from Spirulina spp. biomass in a single-stage extraction with ethanol at room temperature was investigated by Chaiklahan et al. [25]. An increase in the solid to solvent ratio increased the yield of the extracted lipids. A solid: solvent ratio of 1:5 resulted in a lipid yield of 3.6 %, while a solid: solvent ratio of 1:50 resulted in a lipid yield of 7.9 % [25]. The SL and SN percentages are only slightly higher because the Spirulina spp. biomass is a feedstock with low lipid content. The presents study was focused on mathematical numbers obtained from the calculations based on larger-the-better characteristics. According to the purpose of the experiments, the lower-the-better equation is used to minimize the system response (quality decreases as the system response increases), and the larger-the-better equation is used to maximize the system response (quality increases as the system response increases). In contrast, the nominal-the-better equation is used to achieve a particular target value [20].

Point 2: Is Biomass:solvent ratio given as v/v or m/v?

Answer: The biomass: solvent ratio is given as w/v in Table 5.

Point 3: In Conformation experiment, SD value should be provided for the average value of three experiments.

Answer: The SD value (8.7 ± 0.1 %) was added.

Round 2

Reviewer 3 Report

The manuscript was notably improved by authors. I agree with the acceptance of the article. 

I suggest to include in the discussion also the experiments on the temperature and the time of ultrasonication (point 2 and 3). The achieved results are interesting. 

Author Response

Esteemed Reviewer,

Thank you for your very careful review of our paper, and for the comments, corrections and suggestions. Your comments have helped us to carry out a major revision of the paper in order to improve the presentation of the obtained results.

Point 1: The manuscript was notably improved by authors. I agree with the acceptance of the article. I suggest to include in the discussion also the experiments on the temperature and the time of ultrasonication (point 2 and 3). The achieved results are interesting. 

Answer: The results obtained regarding the time of ultrasonication and temperature were added. “Our findings revealed that increasing the ultrasonication time from 5 (25.2 %) to 15 min (25.6 %) leads to a slight increase in the lipid content obtained from Nannochloropsis oculata microalgae mixed with methanol: chloroform (1:2). Preliminary tests showed that an increase in the temperature up to 60 °C led to a slight decrease in the lipid content”.

Reviewer 5 Report

The answers provided by authors did improve the manuscript. The discussion part was substantially improved and minor unclarities were resolved. Therefore, the manuscript can now be accepted for publication.

Author Response

Esteemed Reviewer,

Thank you for your very careful review of our paper, and for the comments, corrections and suggestions. Your comments have helped us to carry out a major revision of the paper in order to improve the presentation of the obtained results.